# Unlocking Full Dynamic Optimization of District Energy Systems through State-Space Model Learning

## Abstract

Predictive control enables the operation of physical systems along an optimal trajectory based on forecasts and dynamic simulations. However, the complexity of system dynamics and high computational cost of optimization typically restrict the optimization window to short horizons. Thus, any potential benefits from mid- and long-term rewards are withdrawn. This is particularly relevant for optimization of district energy systems using various low-environmental-impact sources. To address this, we present an end-to-end methodological framework for learning state-space representations of such systems that significantly reduce computational load. The proposed approach leverages the implicit graph structure of such systems to develop and train a physics-informed spatio-temporal graph neural network. This methodology is evaluated on a real-world district heating system incorporating thermal solar panels, storage, biomass and natural gas boilers. Through historical time-series data augmentation and hyperparameter optimization, the learned model demonstrates strong generalization ability and high accuracy in predicting system dynamics. Our method reduces simulation time by four orders of magnitude, cutting optimization time from several days to mere minutes, while also lowering operational costs by up to 25%.

## 1 Introduction

Mitigating climate change requires substantial reduction in greenhouse gas (GHG) emissions (Portner et al., 2022). To do so, the international energy agency (IEA) outlines the need to deploy large energy networks with multiple low-carbon-footprint energy sources to reach net-zero emissions by 2050 (IEA, 2023). District heating networks are an example of such large energy networks infrastructure (Angelidis et al., 2023). They can use simultaneously various renewable energy sources such as biomass, geothermal, solar thermal, heat pumps in addition to thermal energy storage. Incorporating an increasing number of energy sources requires rethinking smart control strategies to ensure efficient system deployment and achieve sustainable objectives. Nevertheless, the different underlying dynamics (non-linearities, response time, intermittence, discharge rate etc.) brings new complexity to numerical simulation which then makes the optimization of such systems prohibitively time-consuming (Dorotić et al., 2019; Delubac et al., 2021). To tackle this and to reduce the computational load, several approaches have been adopted in literature such as linearization technics (Rojer et al., 2024; Wirtz et al., 2021) and reduced order models (Falay et al., 2020). These approaches often lead to a simplification of the system dynamics and require considerable engineering efforts for each new input variable to the system.

The acceleration of complex and traditional simulations is one of the fields where deep learning models offer an appealing alternative, technically called surrogate models (SM). Neural networks are in general the backbone of these data-based models, thanks to their capacity to capture complex patterns and to handle various data structures (grids, graphs etc.) (Bronstein et al., 2021). This technic was applied to diverse type of dynamical systems such as climate forecasting (Verma et al., 2024), thermal and electrical load forecasting (Wang et al., 2023; Chitalia et al., 2020) and chemical reactors (Ren et al., 2022), among others. More recent theoretical works consider either refining predictions' accuracy (Hua et al., 2023; Beintema et al., 2023) or propose enhanced training procedures to reduce computational resources (Meyer et al., 2023; Fan et al., 2023). Finally, some studies

Figure 1: The proposed methodological framework for application-agnostic predictive control of multi-sources district energy systems. The left block indicates the neural model predictive control scheme in which a validated surrogate model is used along with an evolutionary optimizer. The right figure shows the surrogate model (PI-STGCN) conception, training and validation pipeline.

implemented control strategies where the surrogate model provided fast and accurate prediction of the system response (Jiang et al., 2022; de Jongh et al., 2021)

However, the application of deep learning to physical systems comes with some limitations. In several cases, they are applied on benchmark datasets where data is sampled on small time steps (for example 1ms or 4s) and where the system dynamics relies on few state variables or initial states: a unique inlet velocity value, static motor power for example (Weigand et al., 2023; Pfaff et al., 2020; Schoukens & Noël, 2017). Real-world physical systems are rarely monitored at such time steps and depend on numerous state variables and external inputs (e.g. weather perturbations). Moreover, replacing physical and high-fidelity model with black box neural networks remains an open limitation even with physics-informed models (Cuomo et al., 2022). Furthermore, even though real-world applications are available, no systematic methodology and conception framework have been drawn, specially for district energy networks (Cox et al., 2019; Sun et al., 2022; Yu et al., 2024).

In this work, we propose an end-to-end predictive control methodology for large real-world district energy systems. The proposed approach, schematized in Figure 1, takes advantage of the graph representation of such systems to develop an appropriate physics-informed spatio-temporal graph neural network (PI-STGCN). In contrast to previous works, our proposition is system-agnostic and enables handling various energy sources at different locations along with multiple consumer nodes to learn a state-space representation between these entities. The surrogate model development pipeline, shown in the right block of Figure 1, relies on hyperparameters optimization and historical time-series augmentation used in the learning phase. We demonstrate that, in addition to expanding the dataset size, the latter technique enables the incorporation of physically plausible scenarios into the training set. Our methodology extends beyond the train-validation-test paradigm by further assessing the learned model on unseen data patterns. This extension serves as an additional validation step before using the model with an evolutionary optimization algorithm. The effectiveness of our proposal is demonstrated through its application to a real-world system combining a slow inertial energy source (biomass) with an intermittent source (solar panels).

The primary contributions are summarized as follows:

- We introduce PI-STGCN, a system-agnostic state-space surrogate model for multi-source district energy networks. It enables the modeling of diverse producer and consumer types, accelerating the simulation of these networks. It effectively captures both the fast and slow dynamics of such multi-source systems.

- We propose an adaptation of Gaussian jittering to augment time-series data, exposing the model to plausible training scenarios. The incorporation of first-principle conservation equations allows for more confident predictions. In addition, systematic hyperparameters' optimization is carried out to further enhance the model performance.

- The proposed end-to-end framework bridges the gap between forecasting tasks and predictive control optimization using a state-space surrogate model. This approach aims to accelerate the deployment and management of multi-source energy networks utilizing re-

newable and low-carbon energy sources, contributing to emissions reductions and climate change mitigation.

- We demonstrate the effectiveness of our methodology through its application on a real-world system that uses several energy sources. The choice of this example is based on the heavy constraints involved such as power ramps, minimum time-on/time-off, and minimum technical power. The results show a reduction of operational costs of this system by up to 25% while the computational time was drastically reduced by four orders of magnitude.

## 2 RELATED WORK

Model predictive control as schematized in Figure 1 requires an accurate system model to perform predictive simulations. The control algorithm must accurately model and predict the system's behavior under various control scenarios. In control theory, this dynamical model is often expressed in a state-space where the dynamics follow an ordinary differential equation (ODE) in terms of state variables (Blaud et al., 2023). An optimal control problem is mathematically formulated for time $t \in [0, \ t_f]$ as follows:

$$\frac{dx\,(t)}{dt} = f\left(x(t),\ u(t),\ d(t)\right), \ \text{and} \ x(0) = x_0,$$
$$C\left(t_f,\ u\right) = \int_0^{t_f} g\left(t,\ x_u(t),\ u(t)\right) dt + h(t_f, x_u(t_f)). \tag{1}$$

where $f$ represents the non-linear system dynamics, $x \in \mathbb{R}^{n_x}$ is the vector of state variables, $u \in \mathbb{R}^{n_u}$ and $d \in \mathbb{R}^{n_d}$ are the vectors of control variables and external disturbances respectively. The cost function $C$ is composed of a running cost $g$ and a terminal cost $h$ evaluated at $t = t_f$, the end of the optimization horizon $\mathcal{H}^{opt}$. State-space models can be learned in two distinct ways, discrete-time (DT) or continuous-time (CT) models (Beintema et al., 2023). The latter requires solving an ODE and usually involves initial state estimation (Ayed et al., 2019; Beintema et al., 2023). In contrast, DT models are more common and easier to construct as data is represented via discrete elements (matrices, vectors, etc.).

In the field of district energy systems, a number of studies proposed surrogate DT models (Owerko et al., 2020; de Jongh et al., 2021; Saloux et al., 2023; Boussaid et al., 2024; de Giuli et al., 2024). For example, de Giuli et al. (2024) proposed to associate a recurrent neural network (RNN) to each consumer node in a district heating network (DHN). However, they only considered a single producer network, and the surrogate model conception relies on creating and connecting RNN cells, meaning that GNN could have been used instead. The use of GNN allows encoding topological features of data as inductive bias in the model as in Boussaid et al. (2024). The authors employed a spatio-temporal graph convolution network (GCN) in addition to graph attention (GAT) and proposed a surrogate model to accelerate dynamic simulations by one to two orders of magnitude. However, no physical constraints were incorporated, and similarly, only a single producer networks were presented. Finally, other studies (Huang et al., 2023; Saloux et al., 2023) have proposed control strategies for district energy systems in which heat load forecasts (represented by the blue rectangle in the left block of Figure 1) are generated by data-driven models, while the physical system still used a numerical simulation. Another approach for control of energy systems employs deep reinforcement learning and showcased interesting results (Yeh et al., 2023), but such methods need considerable training times (ranging from several hours to days). The frugality of training our surrogate model is a key advantage and a surrogate model is a highly modular tool, meaning that it can be used either as a predictive model and/or for optimal control. Our work completes the related studies by employing a versatile neural network architecture that is application-agnostic, making it suitable for district energy systems. The model is based on a spatio-temporal graph neural network (Ji et al., 2023) and benefits from recent demonstrations showing that 'time-then-space' models have an expressivity advantage over 'time-and-space' representations (Gao & Ribeiro, 2022). To further enhance generalizability, a physics-informed approach is used in training (Raissi et al., 2019), where a first-principles mass balance constraint, applicable to all district energy systems, is incorporated into the loss function (Guelpa et al., 2019). Finally, the learned model is combined with a genetic algorithm (Deb, 2001) for optimal control of district energy systems.

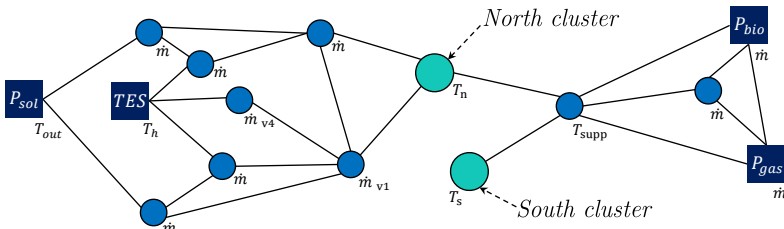

Figure 2: District energy topology studied in this work. The dark blue rectangles represent producers: biomass ($P_{bio}$), natural gas ($P_{gas}$), solar ($P_{sol}$) and storage ($TES$). The small blue circles are control valves (Steiner nodes), while the clear blue circles represent two consumer clusters. The variable under each node corresponds to its associated state variable (node feature).

## 3 METHODOLOGY

### 3.1 PHYSICAL SYSTEM DESCRIPTION

District heating networks consist of several producers delivering heat to consumers via a network of pipes and control valves (equivalent to Steiner points). Therefore, such systems can be suitably represented as a graph $\mathcal{G} = (\mathcal{V}, \mathcal{E})$, where $\mathcal{V}$ is the set of nodes (producer, consumer or valve), and $\mathcal{E}$ the set of edges (i.e., pipes). Section 3.2 will later show that each graph entity holds multiple interconnected physical features. Recent generation of these systems are characterized by employing different producer types at different locations of the network (Pakere et al., 2023). Various types of heat generators can be integrated into district heating networks, including biomass boilers, geothermal sources, natural gas boilers, solar panels, and heat pumps. Thermal energy storage plays a critical role by enabling asynchronous production and peak load shaving. In this work, we consider a real-world DHN featuring three producers: a biomass boiler, a natural gas boiler, and a solar thermal panel field connected to thermal storage. Numerous valves can be seen in Figure 2 between the solar field ($P_{sol}$) and the storage ($TES$), they allow different cycles: Charging or discharging the storage, or direct injection from the solar field to the network. A graph representation of the system is schematized in Figure 2. The considered network incorporates different type of constraints making it a well representative case that requires complex control strategies (Veyron et al., 2022):

- When the biomass boiler is turned on, it must remain so for a minimum time-on $\tau_{min,on}$ and similarly when it is shut down for a minimum time-off $\tau_{min,off}$. Besides, power variations' are limited with ramp constraints.

- Solar energy must be used when available to avoid overheating and to increase the contribution of renewables in the production portfolio. Simultaneously charging and discharging the TES is prohibited.

- Finally, the producers must provide enough heat energy to meet the heat demand of the two clusters while temperature levels must remain above a fixed threshold.

Additional constraints and a detailed mathematical formulation of the physical system are given in appendix A.1. Besides these constraints, network operators depend on external disturbances: solar irradiance $G_{\mathrm{irr}}$ and external temperature $T_{\mathrm{ext}}$, both impact the production of solar energy, and finally the heat demand of the two clusters, $\dot{Q}_{\mathrm{n}}$ (north) and $\dot{Q}_{\mathrm{s}}$ (south). The aim of the predictive control is to optimize the usage of the energy sources (when to switch a source on or off, the power levels of each source, the flow rates, etc.). To do so, network operators dispose of control variables which are the mass flow rates sent to each of the clusters. This updates equation 1 as follows:

$$u(t) = [\dot{m}_{\mathrm{n}}(t),\ \dot{m}_{\mathrm{s}}(t)]\,,\ \text{and}\ d(t) = \left[G_{\mathrm{irr}}(t),\ T_{\mathrm{ext}}(t),\ \dot{Q}_{\mathrm{n}}(t),\ \dot{Q}_{\mathrm{s}}(t)\right]. \tag{2}$$

In such networks, the energy flows at the speed of the fluid in the pipes ($\approx 1 - 2$ m/s), and thermal transients are known to be slow (network time constant $\tau_{\mathrm{network}} \approx$ few hours) (Guelpa et al., 2019). This outlines an important characteristic called 'thermal inertia' or 'distribution phasing'. It means that production at time $t$ arrives to the consumer hours later ($t + \tau_{\mathrm{network}}$), depending on the network size (i.e, pipes lengths) and emphasize the importance of predictive control. The objective of the

predictive control is usually set to minimize the running operating costs (fuel costs), while respecting all the constraints at each time step:

$$C\left(t_f,\, u\right) = \int_0^{t_f} c_{\text{bio}}(t) \cdot \dot{Q}_{\text{bio}}(t) + c_{\text{gas}}(t) \cdot \dot{Q}_{\text{gas}}(t) dt,$$

$$\dot{Q}_i(t) = \dot{m}_i(t) \cdot c_p \cdot \left[ T_{supp}(t) - \frac{\dot{m}_{\text{n}}(t) T_{\text{n}}(t) + \dot{m}_{\text{s}}(t) T_{\text{s}}(t)}{\dot{m}_{\text{n}}(t) + \dot{m}_{\text{s}}(t)} \right]. \tag{3}$$

Where $c_{\text{bio}}$ and $c_{\text{gas}}$ are the specific costs (i.e., in €/kWh) of biomass and gas respectively, $c_p$ is the specific heat capacity of the fluid and $T_{supp}$ the supply temperature provided by the producers. The previous equations (2-3) justify the choice of the state variables (i.e., node features) in Figure 2. The mass flow rates ($\dot{m}$) are the node features for the producers and the control valves. The temperature of the fluid in the pipes ($T$) are the node features for the consumers. The temperature exiting the solar field $T_{out}$ was chosen for the solar field as it is a good representative of the heat absorbed by the fluid. Similarly, the top layer temperature in the storage tank $T_h$ is a good representative of the thermal energy stored in it.

## 3.2 NEURAL PREDICTIVE CONTROL

The finite horizon optimal control expressed in equation 1 aims at providing future trajectory of the system dynamics to optimize given objectives and respect specific constraints. The control problem is solved by minimizing the cost function $C$ over the control variables $u$ for a time horizon $\mathcal{H}$. A time horizon is a predefined period of time, which is a set of consecutive time steps for discrete models. In this work, the system dynamics ($f$ in equation 1) are replaced by a deep learning model $f_\theta$ where $\theta$ are the model weights. As stated in section 3.1, district energy networks are characterized by a significant inertia and producers might have constraints with large temporal durations. In addition to future control signals and forecasted disturbances, the learned model requires access to past observations or measurements of state variables as inputs to accurately predict future system behavior. Equations 1 and 3 translate to a neural predictive control as follows:

$$x_+^{\mathcal{H}^{\text{sm}}} = f_\theta \left( x_-^{\mathcal{H}^{\text{sm}}},\, u_+^{\mathcal{H}^{\text{sm}}},\, d_+^{\mathcal{H}^{\text{sm}}} \right)$$

$$C\left( \mathcal{H}^{\text{opt}},\, u_+^{\mathcal{H}^{\text{opt}}} \right) = \sum_t^{t+\mathcal{H}^{\text{opt}}} \left( c_{\text{bio,t}} \cdot \dot{Q}_{\text{bio,t}} + c_{\text{gas,t}} \cdot \dot{Q}_{\text{gas,t}} \right) \times \Delta t. \tag{4}$$

Two time horizons are defined: $\mathcal{H}^{\text{sm}}$, the predictive range of the surrogate model, and $\mathcal{H}^{\text{opt}}$, the typically longer optimization horizon, requiring autoregressive use of the surrogate model. The subscript $+$ indicate predicted variables, meaning values from the current time $t$ to $t_f = t + \mathcal{H}$. Subscript $-$ indicates past observations or measurement of state variables. This can be rewritten as $x_+^{\mathcal{H}^{\text{sm}}} = [x_t, x_{t+1} \dots, x_{t+\mathcal{H}^{\text{sm}}}]$ and $x_-^{\mathcal{H}^{\text{sm}}} = [x_{t-\mathcal{H}^{\text{sm}}}, \dots, x_{t-2}, x_{t-1}]$. The learned state-space model $f_\theta$ is trained to predict the future states of the system given past observations, future control variables and expected disturbances. The surrogate model weights $\theta$ are optimized with supervised learning from a dataset where the system response (i.e., state variables) to different control variables and disturbances are given. The dataset can either consist of real-world historical data or from a high fidelity numerical simulation.

The model architecture, shown in Figure 3a and developed using `torch-spatiotemporal` library (Cini & Marisca, 2022), is an encoder-processor-decoder configuration where gated recurrent units (GRU) are used for encoding and graph convolution for message passing (Gao & Ribeiro, 2022). State variables $x$ are locally defined, meaning that one state variable is associated to each node. Control variables $u$ and disturbances $d$ are diffused to each node so that the contained information is available to all network components. The figure introduces three hyperparameters that will be optimized: number of GRU layers, the hidden size (HS) and the number of GCN layers.

The model, named PI-STGCN, is trained as the following optimization problem:

$$\underset{\theta}{\text{minimize}} \quad \frac{1}{\text{BS}} \sum_b \frac{1}{\mathcal{H}^{\text{sm}}} \sum_t^{t+\mathcal{H}^{\text{sm}}} \left[ \frac{1}{\mathcal{V}} \sum_n \| \hat{x}_{b,n,t} - x_{b,n,t} \|_2^2 + \lambda \cdot \mathcal{F}_m^2 (u, \hat{x}) \right],$$

$$\text{s.t.} \quad \mathcal{F}_m (u, \hat{x}) = \sum_{\text{producers}} \hat{x}_{b,n,t} - \sum_{\text{consumers}} u_{b,n,t}, \tag{5}$$

$$\hat{x}_+^{\mathcal{H}^{\text{sm}}} = f_\theta \left( x_-^{\mathcal{H}^{\text{sm}}}, u_+^{\mathcal{H}^{\text{sm}}}, d_+^{\mathcal{H}^{\text{sm}}} \right).$$

In equation 5, the loss term is weighted (via $\lambda$) with a physical constraint term represented by $\mathcal{F}_m$. This term is the mass flow rates conservation over the network, where the sum of the flow rates sent to the consumer clusters (i.e., control variables) must be equal to the sum of mass flow rates generated by the producers. The loss is averaged and calculated over a batch of size BS and across all the nodes in the network $\mathcal{V}$. More details about the training pipeline are provided in section 3.3.

Once the surrogate model is trained and considered valid, it is used as the predictive model inside the control loop (Figure 1). The optimization horizon, $\mathcal{H}^{opt}$, is selected based on multiple constraints. First, it is well-established that forecast accuracy for disturbances, such as weather and heat demand, decreases over extended prediction periods. Conversely, setting a shorter optimization horizon may result in the loss of long- and medium-term rewards. In the literature, many related studies have limited their optimization horizons to short periods (ranging from a few hours to a single day) due to the aforementioned reasons, as well as the rising computational costs (Jansen et al., 2024; de Giuli et al., 2024; Jäkle et al., 2023; Wirtz et al., 2021; Quaggiotto et al., 2021). In this work a compromise between these constraints was chosen, we set $\mathcal{H}^{opt} = 1$ week. A genetic algorithm (GA) is then used to generate a population of control signals that will be evaluated using the predictive model. Iteratively, a new population is generated based on the best individuals and genetic combinations of the previous population until convergence to optimal control variables. A detailed presentation of the genetic algorithm implementation using `pymoo` library is provided in appendix A.2 after (Blank & Deb, 2020).

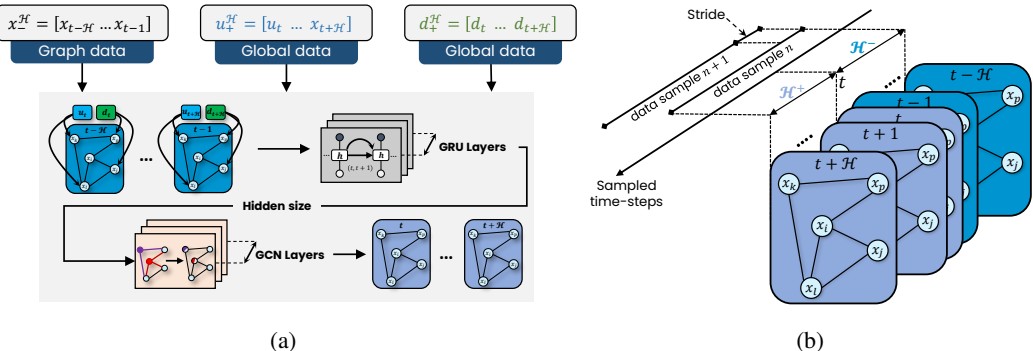

(a)               (b)

Figure 3: Figure (a) illustrates the surrogate model (PI-STGCN) architecture, which integrates past state variables, future control signals, and forecasted disturbances. The number of layers, a key hyperparameter, is optimized (superscript 'sm' is omitted for clarity). Figure (b) depicts the dataset construction process using a sliding window, the impact of the 'Stride' parameter is discussed later. The blue cards are the inputs of the surrogate model and the gray-blue ones are the outputs.

## 3.3 TRAINING PIPELINE

The training and validation pipeline explained in this section is schematized in Figure 1 right block. To construct the dataset, historical measurements are in general available for such systems, specially weather, heat demand and control variables. Therefore, data samples (i.e., $\{x_+^{\mathcal{H}^{\text{sm}}}, x_-^{\mathcal{H}^{\text{sm}}}, u_+^{\mathcal{H}^{\text{sm}}}, d_+^{\mathcal{H}^{\text{sm}}}\}$) are constructed by sliding over the historical data as shown in Figure 3b by a number of time steps called 'stride'. The smallest value for the stride will correspond to the historical data measurement time step. However, this might affect the granularity of the dynamics we want the surrogate model to learn and will be assessed in section 4.2. In general, deep learning

Table 1: Hyperparameters and corresponding search space implemented in ASHA optimizer.

| Hyperparameter | Search space |
|---|---|
| Batch size ($BS$) & Hidden size ($HS$) | $\{64, 128, 256\}$ |
| GRU layers & GCN layers | $\{1, 2, 4, 6, 8\}$ |
| Learning rate ($l_r$) & Physical weight ($\lambda$) | $\left[10^{-4},\ 10^{-1}\right]$ |
| Predictive horizon ($\mathcal{H}^{\mathrm{sm}}$) | $\{12h,\ 24h,\ 48h\}$ |

models are known to require a significant amount of data to effectively learn the desired dynamics. In our case, only one year of historical data was available, to overcome this limitation we propose using time-series augmentation (Nikitin et al., 2023). We implement Gaussian jittering (weak augmentation $\mathcal{T}^1$), where new control and disturbances are generated by using random multipliers $\omega$ and then simulate the system using a high fidelity numerical model. A set of times-series $\omega$ with values in $r = [0.9,\ 1.1]$ is generated via the normal distribution $\mathcal{N}(1, \sigma_{aug})$ where $\sigma_{aug} = (r_{max} - r_{min})/6$. The range $r$ is chosen because the control variables, representing mass flow rates, are constrained by the limited range of hydraulic pumps. However, while Gaussian jittering is commonly used to increase the number of noisy samples, here it is used to generate plausible scenarios. To do so, the sampling frequency ($\Delta t_{aug}$) of random multipliers $\omega$ must be greater than historical data sampling time step $\Delta t_s$. In other words, the data is 'disturbed' every $\Delta = (\Delta t_{aug}/\Delta t_s)$ steps. Let's denote $n$ the number of intervals with $\Delta t_{aug}$ length in the historical dataset, the Gaussian jittering can be formulated as:

$$
\begin{aligned}
\omega_{i|i+\Delta}^{(i,k)} &\sim \mathcal{N}^{(i)}(1, \sigma_{aug}), \quad \text{for } i \in [\![0,\ n]\!] \text{ and } k \in \{u,\ d\}, \\
u_{aug} &= \texttt{Concat}\left(\omega_{i|i+\Delta}^{(i,u)} \odot u_{i|i+\Delta}\right), \quad \text{for } i \in [\![0,\ n]\!], \\
d_{aug} &= \texttt{Concat}\left(\omega_{i|i+\Delta}^{(i,d)} \odot d_{i|i+\Delta}\right), \quad \text{for } i \in [\![0,\ n]\!], \\
x_{aug} &= \texttt{Simulate}(u_{aug}, u_{aug}).
\end{aligned}
\tag{6}
$$

An example of this procedure applied to solar irradiance ($G_{irr}$) is presented in Figure 4a. Two augmented data samples are depicted, illustrating plausible scenarios. The first example represents a sunny day with a brief midday cloud cover, while the second depicts a similar sunny day with slightly higher solar irradiance compared to the original data. An additional illustration is given for one of the control variables ($\dot{m}_{\mathrm{s}}$) and shows the different flow rates generated. The impact of the dataset size (i.e., with and without augmentation) is discussed in the results section.

The dataset is then scaled using min-max normalization and split to three distinct sets, training (70%), validation (10%) and test (20%). The PI-STGCN model is trained using the AdamW (decoupled weight decay regularization) optimizer with a learning rate $l_r$ and a batch size $BS$ (Loshchilov, 2017). To increase model performance, a hyperparameters' optimization is performed using the Asynchronous Successive Halving Algorithm (ASHA) from Li et al. (2018), implemented in `Ray` and `pytorch-lightning` libraries (Liaw et al., 2018; Falcon & team, 2019). The considered hyperparameters and their corresponding range are given in Table 1. The best model is then trained to reach the best optimized results using a 32 GB NVIDIA Tesla V100 GPU. The best model is further evaluated on an additional test dataset of unseen patterns (shown in Figure 4c). The aim of this additional evaluation is to assess the model generalizability to different time-series shapes, potentially resembling those generated by the genetic algorithm.

## 4 EXPERIMENTS

In the following, results are shown for the best model configuration found through hyperparameters' optimization (the five best configurations are given in table 4). The ASHA samples 150 different configuration from the search space specified in Table 1. Unless pruned earlier by the optimizer, each configuration was trained for a maximum of 30 epochs. The averaged mean squared error (MSE) over all the nodes is used as the selection metric (i.e., best model choice is based on it).

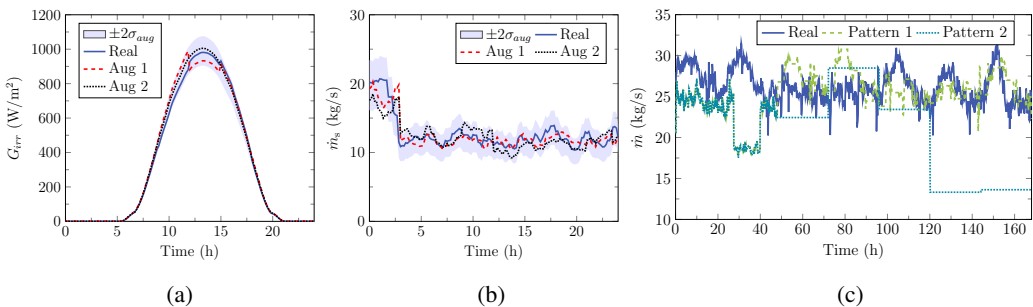

(a)     (b)     (c)

Figure 4: Figures (a) and (b) shows two examples of solar irradiance and control variable time-series augmentation respectively. Figure (c) gives two examples of new control variable patterns used to assess the generalizability of the model.

Besides, back-scaled root mean squared error (RMSE in SI units) is used to measure the model error for variables of interest and particularly the state variables used in the cost function $C$ in equation 3. Finally, the coefficient of determination $R^2$ is used to reflect the model robustness and accuracy. The best model architecture configuration studied here is: $BS = 64$, $HS = 256$, GRU layers $= 1$, GCN layers $= 2$, $l_r = 10^{-3}$, $\lambda = 2.5 \cdot 10^{-4}$ and $\mathcal{H}^{\text{sm}} = 12h$. The stride hyperparameter is set to the smallest possible value, stride $= 10$ min. The impact of choosing bigger strides is discussed later in this section. In the following subsection, model performance is analyzed through error analysis and the effect of Gaussian jittering is discussed.

## 4.1 Model performance

The model predictions are compared to outputs from the numerical simulation to be substituted. The latter comes from a high fidelity and previously validated numerical model implemented in `Dymola` software. An example of PI-STGCN predictions on a test set batch are shown in Figure 5.

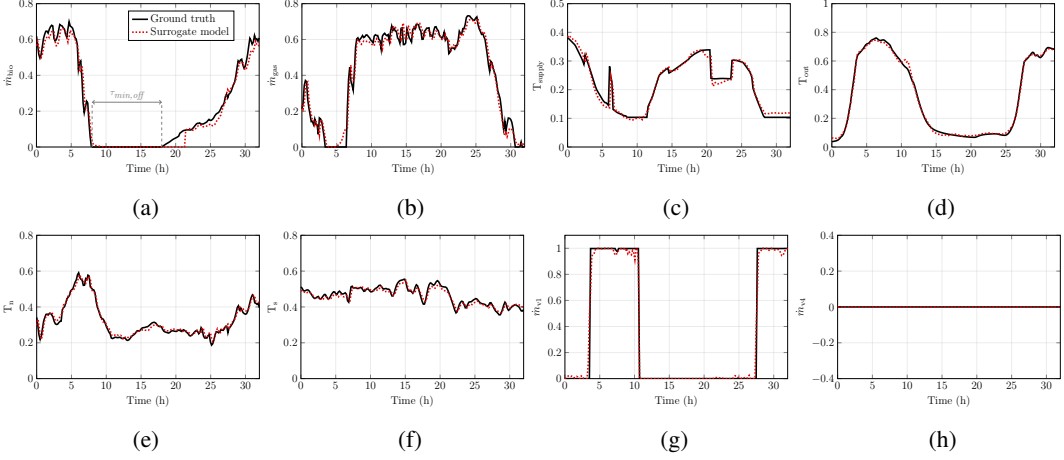

(a)   (b)   (c)   (d)

(e)   (f)   (g)   (h)

Figure 5: Normalized time-series results and comparison between PI-STGCN predictions (red dotted curve) and simulation (black curve). The different dynamics are well captured and the predictions errors are in an acceptable range for network operators. Only variables used in the cost function are shown here, additional state variables predictions are illustrated in appendix A.3.

It can be seen that different dynamic patterns are well captured, both fast (5b) and relatively slow (5d) evolutions are learned. Moreover, the on/off behavior of control valves (5g, 5h) is precisely learned, this makes the model remarkably accurate. The minimum time-off constraint ($\tau_{min,off}$) remains respected for the biomass boiler as shown in Figure 5a. These results are obtained using a dataset of three years simulation, two of which are generated by the historical data augmentation presented previously. Table 2 presents error values, performance metrics, and evaluates the impact

Table 2: The best model configuration performance assessment through various metrics.

| Metrics | MSE - | $R^2$ - | $\mathcal{F}_m$ - | $\dot{m}_{bio}$ (kg/s) | $\dot{m}_{gas}$ (kg/s) | $T_{supp}$ (K) | $T_n$ (K) | $T_s$ (K) | Error reduc. | Training time |
|---|---|---|---|---|---|---|---|---|---|---|
| Data augmentation assessment | | | | | | | | | | |
| **1RY** | 0.024 | 0.77 | 0.042 | 3.7 | 4.2 | 1.2 | 2.6 | 2.4 | Ref | 33 min |
| **1RY+1AY** | 0.006 | 0.96 | 0.031 | 3.2 | 3.5 | 1.2 | 4.3 | 1.9 | 75% | 1h 25 min |
| **1RY+2AY** | 0.004 | 0.99 | 0.019 | 2.7 | 3.2 | 0.8 | 1.4 | 1.4 | 83% | 1h 56 min |
| Evaluation on new patterns in Figure 4c | | | | | | | | | | |
| **Pattern 1** | 0.001 | 0.98 | 0.002 | 1.3 | 1.6 | 0.4 | 0.9 | 1.1 | - | - |
| **Pattern 2** | 0.002 | 0.98 | 0.016 | 2.5 | 2.9 | 0.5 | 2.4 | 1.7 | - | - |
| Impact of stride hyperparameter mentioned in Figure 3b | | | | | | | | | | |
| **S = 1 h** | 0.007 | 0.94 | 0.017 | 3.5 | 4.1 | 1.2 | 2.2 | 2.3 | Ref | 19 min |
| **S = 30 min** | 0.006 | 0.97 | 0.016 | 3.3 | 3.9 | 1.1 | 2.0 | 2.1 | 14% | 40 min |
| **S = 10 min** | 0.004 | 0.99 | 0.019 | 2.7 | 3.2 | 0.8 | 1.4 | 1.4 | 42% | 1h 56 min |

of data augmentation. The acronyms RY and AY stand for 'Real Year' and 'Augmented Year', respectively. The optimal model configuration was trained on three datasets (1RY, 1RY+1AY, and 1RY+2AY) to achieve peak performance. All models were tested on the same dataset, covering 7 months of data (late summer, autumn, and early winter). The best performance was observed when the model was trained using a combination of one real year and two augmented years (1RY+2AY). The performance enhancement is significant as the normalized MSE is reduced by over 83% compared to the model trained with 1RY dataset. The $R^2$ value is also improved, indicating a better fit of the model. Besides, the RMSE of state variables used for calculating the cost function $C$ in equation 3 are given, and are in acceptable range for network operators. Finally, such high accuracy comes also with a drastic decrease of four orders of magnitude (reduction factor = $1.9 \cdot 10^4$) in computational time.

## 4.2 DYNAMIC EFFECTS

In this section two aspects of the model performance are analyzed. First, the generalizability of the surrogate model is assessed by measuring its accuracy for two weeks of simulation where control variables follow different patterns from the one in the training dataset (shown in Figure 4c). The results are reported in Table 2. MSE values indicate that the model predictions are notably accurate and confirm that the model effectively learned the underlying dynamics of the studied system. In terms of comparison, the accuracy is slightly lower for 'Pattern 2' as expected. In fact, this signal is made up of successive long-term trays, a feature completely unavailable in the training dataset. Therefore, the model effective and accurate performance (i.e., no significant degradation) is confirmed and make it now available for using it in a control loop.

The final aspect addresses the influence of the 'stride' hyperparameter during dataset construction, shown in Figure 3b. Three stride values were tested using the optimal model configuration and the 1RY+2AY dataset. The smallest stride corresponds to the real sampling frequency (weather data is available every 10 minutes), the same as the frequency used in numerical simulations. Results indicate that model performance improves as the stride decreases, the error is 42% lower when using s = 10 min instead of s = 1 h. A smaller stride increases the number of samples in the training set for a given dataset size, but more crucially, it enhances the model's ability to capture rapid dynamics, particularly mass flow rates.

## 4.3 APPLICATION TO OPTIMAL CONTROL

After validating the learned state-space model, we provide a demonstration of how it can be used in predictive optimal control of district energy systems. The methodology as illustrated in Figure 1 relies on using the surrogate model to provide objective function estimates for the different control variable scenarios generated by the optimizer (genetic algorithm in this case).

Table 3: Optimization results using the surrogate model (NPC) compared to actual costs, with computational time comparison if the numerical model (MPC) is used.

|  | Real costs (k€) | Optimized costs (k€) | Cost reduction | MPC comp. time | NPC comp. time | Time reduction |
|---|---|---|---|---|---|---|
| **Week 1** | 49.7 | 45.7 | 8% | $\sim$ 6 days | $\sim$ 8 min | $\sim 1.1 \cdot 10^3$ |
| **Week 2** | 30.6 | 26.5 | 13% | $\sim$ 8 days | $\sim$ 9 min | $\sim 1.3 \cdot 10^3$ |
| **Week 3** | 19.4 | 14.5 | 25% | $\sim$ 13 days | $\sim$ 10 min | $\sim 2 \cdot 10^3$ |

As a proof of concept, we consider an optimization horizon of 1 week and retrieve three representative weeks from historical data where real costs are available:

- **Week 1:** week with the highest total heat load, it occurs during winter ($8^{th}$ to $15^{th}$ of December), with cold weather conditions, low irradiance and high flow rates required.

- **Week 2:** week with the median total heat load, it occurs during spring ($12^{th}$ to $19^{th}$ of May), with variable irradiance and a highly fluctuating load.

- **Week 3:** week with the lowest total heat load, it occurs during summer ($28^{th}$ of July to $4^{th}$ of August), with high irradiance during the day, the thermal storage will be heavily used.

The objective function corresponds to the running cost $C$ defined in equation 3. Interestingly, by learning a state-space constrained model, the optimal control problem becomes an unconstrained problem. In fact, the numerical simulation model already incorporates the different constraints presented in section 3.1, meaning that the outputs of the surrogate model are implicitly constrained. Once the genetic algorithm reaches convergence, the optimal solution found by the surrogate model is sent to the system, i.e. the high fidelity numerical model to confirm its feasibility and optimal cost. The GA algorithm is executed with a population of 100 individuals (one-week time series for each control variable) evolving over 200 generations. The results, summarized in Table 3, indicate significant cost reductions compared to real system operations. The highest reduction, 25%, occurs in week 3, where optimal use of solar power and storage minimizes costs. In contrast, for week 1, which falls in winter with the highest heat demand, operational costs are reduced by 8%. Due to low solar availability and demand constraints, high mass flow rates are necessary, resulting in biomass and gad boilers operating at high load. An intermediate cost reduction of 13% is observed for the median week.

One of the most notable results is the significant reduction in computation time. Using the GA optimizer with the `Dymola` software requires several days of calculation, rendering it infeasible for weeks 2 and 3, where the optimization calculation time exceeds its predictive horizon. By contrast, the inference model reduces the computational time to just a few minutes. The variation in computation times across weeks is attributed to the complexity of the system's dynamics. Week 3, in summer, requires more time due to the involvement of thermal storage, with multiple cycles (charging, direct injection, discharging, etc.). This drastic improvement enables real-time use of the optimal control framework and highlights the clear advantages of deep learning models in promoting, optimizing and deploying district energy systems effectively.

## 5 CONCLUSION

In this work, we presented a streamlined methodology for deep learning-based optimal control for computationally intensive multi-source district energy networks simulations. A comprehensive framework for training and validating the physically-informed surrogate model is provided. It relies on historical data augmentation through Gaussian jittering and hyperparameters optimization. The learned model is then used by a genetic algorithm optimizer to provide accurate estimates to optimize a given objective function. The results demonstrate the effectiveness of our approach in optimizing costs (up to 25%) and reducing computational time (from several days to few minutes). This work opens up new perspectives for the optimized deployment and control of multi-source district energy systems, thus contributing to the decarbonization of energy systems to meet environmental objectives.

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

# A APPENDIX

## A.1 PHYSICAL CONSTRAINTS

As stated in section 3.1, the study case incorporates three energy sources: biomass, natural gas and solar thermal panels. The latter is also connected to a thermal energy storage system (TES). The first physical constraint that must be met by the system is the energy balance that ensures the consumers' heat demand is satisfied at every time step $t$:

$$\dot{Q}_{\text{sol,t}}^{nw} + x_{\text{bio,t}} \dot{Q}_{\text{bio}}^{min} + \dot{Q}_{\text{bio,t}} + \dot{Q}_{\text{gas,t}} + \dot{Q}_{\text{TES,t}} = \dot{Q}_{\text{n,t}} + \dot{Q}_{\text{s,t}}. \tag{7}$$

Where $x_{\text{bio}}$ is a binary variable that controls the On/Off status of the biomass boiler. In addition, the latter can provide energy to the network only from a minimum technical power $\dot{Q}_{\text{bio}}^{min}$. Concerning the solar power, it can be seen that the corresponding term has a superscript $nw$, referring to direct injection. In fact, the solar power can be either stored (superscript $st$) or used directly to heat the network via the following energy balance:

$$\frac{\dot{Q}_{\text{sol,t}}^{nw} + \dot{Q}_{\text{sol,t}}^{st}}{A_{\text{sol}}} = \eta_0 G_{irr,t} - a_1 \left(T_{\text{fluid,t}} - T_{ext,t}\right) - a_2 \left(T_{\text{fluid,t}} - T_{ext,t}\right)^2 - c_{eff} \frac{dT_{\text{fluid}}}{dt} \tag{8}$$

Where $A_{\text{sol}}$ is the total surface of the solar thermal panels. The parameters $\eta_0$, $a_1$, $a_2$ and $c_{eff}$ are generally taken from the constructor technical sheets. This equation is non-linear given the temperature difference term associated with coefficient $a_2$.

The biomass boiler has significant inertial constraints such as minimum time-on and time-off and ramp constraints. Such constraints are mathematically formulated as:

$$x_{\text{bio,t}} - x_{\text{bio,t-1}} \geq -1 + \sum_{k=t}^{\min(t+\tau_{min,off}, \mathcal{H})} \frac{x_{\text{bio,k}}}{\tau_{min,off}}, \tag{9}$$

$$x_{\text{bio,t-1}} - x_{\text{bio,t}} \leq \sum_{k=\max(0, t-\tau_{min,on})}^{i} \frac{x_{\text{bio,k}}}{\tau_{min,on}}. \tag{10}$$

$$\dot{Q}_{\text{bio,t}} - \dot{Q}_{\text{bio,t-1}} \leq r_{up} \dot{Q}_{\text{bio}}^{max}, \tag{11}$$

$$\dot{Q}_{\text{bio,t}} - \dot{Q}_{\text{bio,t-1}} \geq -r_{down} \dot{Q}_{\text{bio}}^{max}, \tag{12}$$

$$\dot{Q}_{\text{bio,t}} \leq x_{\text{bio,t}} \left(\dot{Q}_{\text{bio}}^{max} - \dot{Q}_{\text{bio}}^{min}\right). \tag{13}$$

Equations (9, 10) ensure the minimum time-on and time-off constraints are respected. The following two equations (11, 12) control power variations as stiff variations could damage the boiler. The last equation ensures that the biomass boiler is only available when it is turned on. The thermal energy storage physical model is a stratified layers model similar to the one developed in Untrau et al. (2023). Finally, the charged (from solar) and discharged (to the network) quantities must respect the maximum capacity of the storage, and simultaneous charging and discharging are prohibited.

$$\dot{Q}_{\text{sol,t}}^{st} \leq x_{st,t}^c \frac{C_{st}^{max} - C_{st,t-1}}{\Delta t}, \tag{14}$$

$$\dot{Q}_{\text{TES,t}} \leq x_{st,t}^d \frac{C_{st,t-1}}{\Delta t}, \tag{15}$$

$$x_{st,t}^c + x_{st,t}^d \leq 1. \tag{16}$$

Equation 16 prevents the storage from charging and discharging simultaneously. The remaining inequalities (14, 15) define the capacity limits for charging and discharging, respectively. Finally, the supply temperature delivered by the boilers follows a 'water-law'.

$$T_{supp} = \begin{cases} 95\,°\text{C} & \text{if } T_{ext} < 0, \\ -0.73 \cdot T_{ext} + 95 & \text{if } T_{ext} \in [0; 15], \\ 84 & \text{if } T_{ext} > 15. \end{cases} \tag{17}$$

## A.2 GENETIC ALGORITHM

We implement a genetic algorithm using `pymoo` library (Blank & Deb, 2020). It offers several evolutionary algorithms that can handle a single or multi-objectives optimization problems. In this work, we only considered the economic cost as the optimization objective, which is technically called the 'fitness function' in the field of evolutionary algorithms. The genetic algorithm optimizer

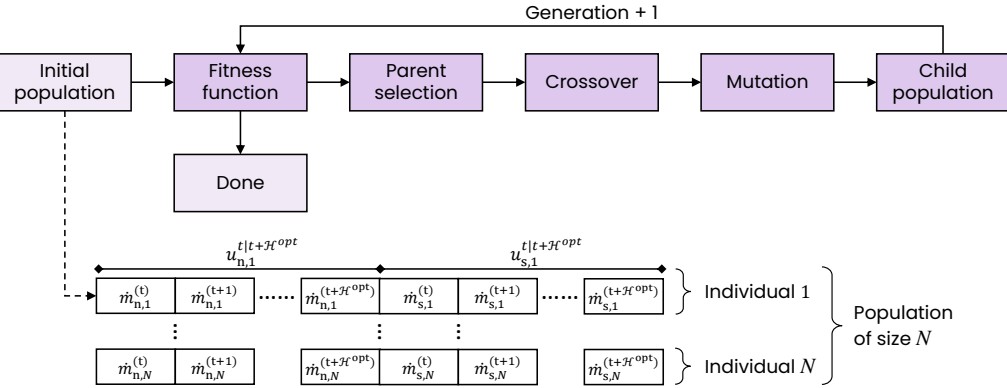

Figure 6: Genetic algorithm optimizer framework. The color of the boxes is relative to the optimizer color shown in Figure 1.

framework is shown in Figure 6. In this work, an individual, i.e., a possible solution to the optimization problem, consists of two time-series with equal length each, which is the optimization predictive horizon $\mathcal{H}^{opt}$. The first is the mass flow rates sent to the north cluster and the second to the south cluster. The crossover operation generates a new individual by combining segments from two parent individuals. In this study, we employ a four-point crossover, where the offspring is formed by selecting two segments from each parent. The arrangement of these segments is randomly determined for each new individual. A polynomial mutation is used to randomly change the position of two time step values in each individual. More theoretical details along with software implementation can be found in Deb et al. (2007). This iterative process is run iteratively until a predefined exit criteria is met. In our case, each optimization was run for 200 generations with a population size of 100. These are empirical values found to be enough for the optimizer to converge to an optimal cost value.

## A.3 MODEL PERFORMANCE

In figure 7, we show three additional state-variables: $\dot{m}_{v6}$ and $\dot{m}_{v2}$ which are two control valves associated with the solar field. The latter is activated similarly to $\dot{m}_{v1}$ in Figure 5g, while $\dot{m}_{v6}$ remain inactive. The fluid temperature in the storage 7b remains constant, meaning the storage is not used.

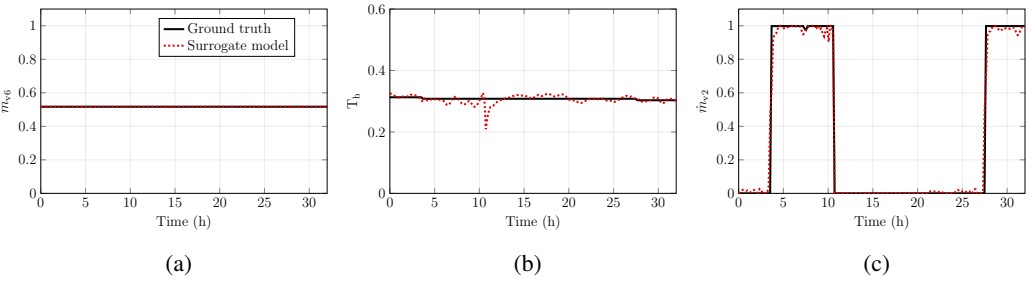

Figure 7: Additional state variables predictions. The model is able to predict zero and non-zeros constant variables.

Table 4: Top five best models hyperparameters configurations, found by the ASHA optimizer.

| Model rank | MSE | $\mathcal{F}_m$ | GRU layers | GCN layers | $l_r$ | $\lambda$ | $\mathcal{H}^{\mathrm{sm}}$ | $BS$ | $HS$ |
|---|---|---|---|---|---|---|---|---|---|
| **1** | $7.78\,10^{-3}$ | $4.7\,10^{-2}$ | 1 | 2 | $1.0\,10^{-3}$ | $2.5\,10^{-4}$ | $12\,h$ | 64 | 256 |
| **2** | $7.99\,10^{-3}$ | $5.6\,10^{-2}$ | 8 | 2 | $1.1\,10^{-3}$ | $5.2\,10^{-4}$ | $12\,h$ | 128 | 256 |
| **3** | $8.07\,10^{-3}$ | $3.8\,10^{-2}$ | 6 | 4 | $9.0\,10^{-4}$ | $4.4\,10^{-3}$ | $12\,h$ | 64 | 128 |
| **4** | $8.15\,10^{-3}$ | $3.2\,10^{-2}$ | 6 | 1 | $6.9\,10^{-4}$ | $4.2\,10^{-3}$ | $12\,h$ | 64 | 256 |
| **5** | $8.17\,10^{-3}$ | $3.5\,10^{-2}$ | 6 | 4 | $3.9\,10^{-4}$ | $5.5\,10^{-3}$ | $12\,h$ | 64 | 256 |

