# OpenReview forum: "Unlocking Full Dynamic Optimization of District Energy Systems through State-Space Model Learning"
_ICLR.cc/2025/Conference — Submitted to ICLR 2025_

### Official Review · Reviewer_zFzQ · 2024-11-02

**Soundness:** 2
**Presentation:** 2
**Contribution:** 2
**Rating:** 5
**Confidence:** 3

**Summary:**

The authors propose an end-to-end framework for optimizing the control of district energy systems. Traditional optimization in such systems is computationally intensive, often limiting the optimization to short horizons and preventing effective long-term planning. The paper addresses this by introducing a physics-informed spatio-temporal graph neural network, which learns a state-space representation of the system dynamics. This surrogate model is optimized to replace high-fidelity simulations, which allows for a significant reduction of computational time and operating costs. The model is applied to a real-world district heating system involving renewable and traditional energy sources such as biomass and natural gas.

**Strengths:**

Using a physics informed, space temporal graph neural network to learn a state space representation of the dynamics in a multi-source district energy networks, to the best of my knowledge, is novel and allows for a significant speed up over model predictive control and reduction of operational costs.

**Weaknesses:**

Since the predictive model is pretained, the overall performance of the proposed method relies on the accuracy of its prediction. Given that the training of this model tries to account for the possible predicted control actions and dynamic scenarios, what if this assumption does not hold, in practice? Alternatively, would it be possible to learn end-to-end the predictive model? That might allow to account for the decision making part. If not, what are the limitations?

**Questions:**

Being a little unfamiliar with the topic, I would like to improve my understanding of the paper to provide better feedback and evaluation.

The system dynamics are modeled using a PI-STGCN; is this surrogate model learned in the loop? If so, how does it account for the control action optimized by the genetic algorithm? More formally, how are the gradients of the loss with respect to the model parameters computed in an end-to-end fashion?
Intuitively, to output good control actions, the genetic algorithm should be informed from the resulting outcome of its predicted control action. Is that correct? Can you provide a step-by-step explanation of how the surrogate model interacts with the genetic algorithm during training and optimization?
What is the size of the actual energy network used in the experiments (number of nodes and edge)? Have the experiments been carried out over different networks size? If so, what is the relationship with the dimensionality of the state-space model that is learned?

---

> ### Author Response · Authors · 2024-11-25
> **Answers to Reviewer zFzQ**
>
> *The authors express their sincere thanks to the reviewer for providing this valuable feedback which will play a crucial role in improving this manuscript in future versions.* Given all the concording reviews we received, the paper seems to need undergoing major changes which we are unfortunately unable to perform and process to update the paper in the given time.
>
> We did try however to answer some of the questions where our choices are justifiable:
>
> **(a) Is this surrogate model learned in the loop?**
>
> The surrogate model is first learned and validated using historical data then called by the surrogate model. The latter is used in inference mode to provide the genetic algorithm with states evolution given the control signals constructed by the genetic algorithm.
>
> **(b) What is the size of the actual energy network used in the experiments (number of nodes and edge)?**
>
> The actual energy network serves 1500 consumer nodes and the total length of the edges is around 10 km. However, to simplify the physical model, the 1500 consumer nodes are separated into two distinct clusters based on the localization of the node (north cluster and south cluster).
>
> **(c) Have the experiments been carried out over different networks size? If so, what is the relationship with the dimensionality of the state-space model that is learned?**
>
> Unfortunately, such tests require the access to real data from different networks which is very complicated as these data are very often confidential and private. But we can eventually perform it on “fictional” networks for the part of learning the dynamics only.
>
> Best regards.

---

### Official Review · Reviewer_eNGU · 2024-11-07

**Soundness:** 2
**Presentation:** 3
**Contribution:** 3
**Rating:** 3
**Confidence:** 4

**Summary:**

This paper presents a framework for learning state-space representations, that leverages graph neural networks to lead to computational gains. The method is evaluated on prediction and control scenarios. While the paper was overall well written, it contains some misleading comparisons and omissions that make it not good enough for publication at this stage.

**Strengths:**

1. historical timeseries augmentation is a very clever idea, and the combination of using a high fidelity surrogate model, combined with gaussian jittering, to augment the data is elegant and well presented.

2. While the authors are not the first to use a GNN for this problem, the choice is well motivated and clearly appropriate.

3. Equations were neatly presented, and I appreciated the level of detail.

4. The approach is system agnostic

5. The approach is clearly presented and its novelty relative to prior work is mostly well explained.

**Weaknesses:**

1. While the writing quality was mostly good, there were some issues that need to be resolved. For example, some typos persist throughout the paper, such as the misspelling of “technique” as “technic” on lines 42 and 50, and confusing “optimized” and “optimal” on line 512.
There were also several examples of awkward writing, such as
\
“The acceleration of complex and traditional simulations is one of the fields where deep learning models offer an appealing alternative, technically called surrogate models (SM)”
\
“A detailed presentation of the genetic algorithm implementation using pymoo library is provided in appendix A.2 after”
\
The writing was also hyperbolic in some cases, such as
“Our work completes the related studies by employing a versatile neural network architecture that is application-agnostic, making it suitable for district energy systems.”
\
Another issue is that some terms are not defined. For example, the authors referenced “weak augmentation T_1” on line 337, but I could not find this defined anywhere, neither in the paper nor in the reference it cited.

2. Assuming we have an accurate predictive model, genetic algorithms are a strange choice of baseline. The results were good enough that I am not denying the efficacy of genetic algorithms, but there needs to be more motivation behind why this was chosen as opposed to other MPC or RL control algorithms.

3. Claims of the speedup relative to other systems as “4 orders of magnitutde” seems wildly exaggerated for the following reasons:
\
a. Their model is being compared with a “high fidelity numerical model”. However, we are not given details of this model at all! This is a glaring lacuna. Further, we do not know what hardware it is run. Conversely, their alg makes use of a V100 GPU, which is a significant amount of compute
\
b. Their model required hyperparameter tuning, with 150 sampled configurations with 30 epochs each. This cost is simply ignored in the comparison
\
c. The selfsame expensive “high fidelity numerical model” is used in the data augmentation pipeline. This cost is completely omitted as well.

4. The fact that the “high fidelity numerical model”, also referred to at times as the “simulator” is not clearly defined is a glaring omission. Not just because this makes computational comparisons impossible to verify, as discussed above, but also because this model is used in the data augmentation pipeline. Furthermore, if this highly accurate model exists, why do we not have a baseline comparing performance gains when we use it for MPC? In this regard, Table 3 is highly misleading. It compares performance on real data vs their method, but computation is a comparison of their method vs using MPC. Surprisingly missing is the performance of the MPC model!

5. The authors claim, as early as the abstract, that “This methodology is evaluated on a real-world district heating system incorporating thermal solar panels, storage, biomass and natural gas boilers.” Initially, this claim really excited me, but as I read the paper I found it highly misleading as well.
\
This implies a live trial on a real system. However, what was actually performed was a comparison of real offline data, and the performance of their model on a high fidelity simulator. This is decidedly NOT a real world evaluation. Furthermore, since the very evaluation model, the high fidelity numerical model, is also used in the augmentation pipeline, this is a serious evaluation issue. For example, in section 4.1 the model is compared with the simulator, but since it was trained on augmented data from the simulator, this is equivalent to showing performance on the training set.

**Questions:**

I would like clarification on the following questions:

1. What are the details of the simulator used?

2. What is the performance of MPC using the simulator itself for rollouts?

3. What computing power was used for the simulator, so that we can actually understand the proposed speedup

4. How does the hyper param tuning and data augmentation factor in to the speedup? We cannot ignore this, even if the actual simulation time once it is done is much faster

5. Did you use the same simulator both to augment the dataset, and to evaluate the model? Can you provide clarification on why this is not simply overfitting to the simulator?

---

> ### Author Response · Authors · 2024-11-25
> **Answers to Reviewer eNGU**
>
> *The authors express their gratitude to the reviewer for his valuable review and constructive comments.* Given all the concording reviews we received, the paper seems to need undergoing major changes which we are unfortunately unable to perform and process to update the paper in the given time.
>
> We did try however to answer some of the questions where our choices are justifiable:
>
> **(1) What are the details of the simulator used?**
>
> The ground truth physical model was indeed very briefly mentioned and a reference for a detailed explanation of the model was given. Given the details of it would require another standalone paper, is eventually not of big interest given the conference topics. We deliberately chose not to explain every part of it given that we provided a reference where this model is presented, detailed and validated through comparison with real data, and because of the pages limit.
>
> **(2) What is the performance of MPC using the simulator itself for rollouts?**
>
> This was not given as we only wanted to compare the equivalent time that would have taken the MPC to run the same number of iterations using the physical simulator (much slower than the surrogate model). However, we understand why it is relevant to provide it and will add it in our further research.
>
> **(3) What computing power was used for the simulator, so that we can actually understand the proposed speedup.
> (4) How does the hyper param tuning and data augmentation factor in to the speedup? We cannot ignore this, even if the actual simulation time once it is done is much faster.**
>
> The proposed speedup comes essentially from the fact that the surrogate model is a forward function and requires no iterations and/or solving non-linear equations which is the case of the physical simulator. However, we understand the comment about taking into account the complete training/hyperparameter tuning time when evaluating the computational load that was necessary to construct an accurate surrogate model.
>
> **(5) Did you use the same simulator both to augment the dataset, and to evaluate the model? Can you provide clarification on why this is not simply overfitting to the simulator?**
>
> We do not want to overfit the simulator but to imitate it, as it is a high-fidelity model of the real functioning of the district energy system studied. The augmented dataset was firstly generated by the simulator then the surrogate model was trained on this dataset. The goal was to increase the size of the dataset as the available data (only one year) was not sufficient to achieve high performance with the surrogate model.
>
> Best regards.

---

> > ### Comment · Reviewer_eNGU · 2024-11-26
> > **Response to Authors**
> >
> > Thank you for all your clarifying comments. Wishing you all the best of luck in the future, and I am sure with some revisions this will be a great contribution.

---

### Official Review · Reviewer_e7E6 · 2024-11-08

**Soundness:** 2
**Presentation:** 2
**Contribution:** 3
**Rating:** 3
**Confidence:** 4

**Summary:**

This paper addresses the control of district energy systems using a neural network surrogate model for the dynamics; the aim is to improve control performance while reducing computational costs. The main contribution is PI-STGCN, a neural network-based model that is trained to approximate the dynamics by minimizing a supervised loss with an additional soft constraint representing mass flow rate conservation. The authors present results demonstrating that this surrogate model captures both fast and slow dynamical patterns within the system. PI-STGCN is then used within an optimal control loop, which is optimized using genetic algorithms. The authors show that this approach improves system costs compared to the business-as-usual system operation, while being significantly faster to run than MPC.

**Strengths:**

* This paper tackles a problem of great societal importance -- namely, optimization of district heating networks.
* The need for a surrogate model to approximate the system dynamics is well-motivated, and the use of a supervised loss plus mass flow rate conservation term is well-suited for this problem. The surrogate model seems to perform well, capturing both fast-evolving and slow-evolving dynamics that are apparent in the numerical simulation.
* The proposed method improves control performance by ~8-25% compared to business-as-usual operations (depending on which scenario), and is orders of magnitude faster to run than MPC.

**Weaknesses:**

* The experimental evaluation needs significant improvement:
   * With respect to learning the dynamics model, no other methods are compared against; all comparisons depict different configurations of the authors' proposed model. There should be a more extensive comparison against baselines and alternative methods.
   * The comparisons with respect to control performance seem incomplete. In particular, there is little transparency on what control method is used to generate the "real costs." In addition, while timing performance is shown for MPC, the control costs are not shown for MPC. More extensive comparison against baselines and alternative methods is likewise necessary.
   * Ablation studies are needed to demonstrate that all parts of the proposed algorithmic pipeline are indeed necessary, as there are many moving pieces to the proposed methodology.
* The writing and presentation could use some improvement. This includes:
    * Ensuring all terms are defined in the main body of the paper, alongside where they are used.
    * Ensuring the main body of the paper is self-contained with respect to the methodology. Notably, details on the genetic algorithm-based optimization of the control loop are relatively sparse.
    * Improving the discussion of prior work. Despite the existing discussion of prior work, it is not clear what exactly is the state-of-the-art upon which this paper improves (and this is further exacerbated by the lack of comparisons in the experiments).

I also have several questions, which may or may not indicate weaknesses:
* A Gaussian jittering approach may lead to time series that have the exact "convenient" type of noise/deviation from the original signal that it is easy for neural networks to smooth out. Is the evaluation in Figure 5 on those same Gaussian-jittered time series on which the model was trained, or on a different "hold out" set of time series?
* How "good" are the MSEs in Table 2 (relating to out-of-distribution dynamics)? Is it possible to see a plot in order to visually inspect how well the out-of-distribution dynamics were learned?
* In Section 4.3, it is claimed that by learning a neural network surrogate for the dynamics, it is no longer necessary to view the control problem as a constrained problem. Could the authors provide more substantiation for this claim? Is this claim solely based on the observation that constraints were satisfied in Figure 5 (evaluation of the dynamics learned by PI-STGCN), or is there theoretical proof? If the former, then it would be seriously important to also report on constraint satisfaction performance in Section 4.3 / Table 5 (evaluation of control performance).

**Questions:**

* How do stronger baselines and ablations of the proposed method perform in this setting? (See more detail in "Weaknesses.")
* What is the control cost associated with using MPC?
* See also the additional questions under "Weaknesses."

---

> ### Author Response · Authors · 2024-11-25
> **Answers to Reviewer e7E6**
>
> *The authors express their sincere thanks to the reviewer for providing this valuable feedback which will play a crucial role in improving this manuscript in future versions.*
> Given all the concording reviews we received, the paper seems to need undergoing major changes which we are unfortunately unable to perform and process to update the paper in the given time.
>
> We did try however to answer some of the questions where our choices are justifiable:
>
> **(a) How do stronger baselines and ablations of the proposed method perform in this setting? (See more detail in "Weaknesses.")**
>
> The comparison with baselines for learning dynamics were not given as we thought that it has been sufficiently shown in the literature that state-of-the-art results for predicting time-series on graphs are obtained using spatiotemporal graph neural networks. However, we will include them in our future research as it is often appreciated to provide some “experimental” proof of it.
> We will also give more experimental details about the comparison between control methods and the resulting costs (including MPC).
>
> Best regards.

---

### Official Review · Reviewer_XnjS · 2024-11-10

**Soundness:** 2
**Presentation:** 2
**Contribution:** 1
**Rating:** 3
**Confidence:** 3

**Summary:**

This paper looks into the long-horizon predictive control problem of district heating networks. A state space model is formulated, and a genetic algorithm based controller is adopted. Simulation results show the proposed method can help with both computation time and solution accuracy.

**Strengths:**

This paper brings out the modeling and control of district heating network. It brings the strong modeling power of graph neural networks into this application. The real data also justifies the use of machine learning model for such a task.

**Weaknesses:**

One of the major concerns are the gap between a general "application of deep learning to physical systems" and the specific "district energy systems" tasks. The authors made several bold claims about the method, but the major contribution seems to be limited to a very specific physical dynamics and control settings.

Another major concern is the use of genetic algorithm. It is unclear why such algorithm is selected, and how the algorithm performs with model-based RL or model-free decision-making approaches.

The use of STGCN in dynamics modeling and decision making are not novel contributions of this work. It is a mere application of the previous packages.

The abstract described about long-term reward, but it is unclear how the proposed method tackle such issue.

There is a lack of comparison to other methods, and the testing case seem to be very specific to one private dataset.

Some minor errors:
Line 209 "each of the clusters"-> "each of the cluster";

**Questions:**

(a). What do RY, AY mean in the dataset? Needs a bit more explanation.

(b). For the real data mentioned in the paper, is it possible to get the ground truth physical model? And based on such a model, can standard Model Predictive Control or optimization-based methods give exact solutions? I notice the authors compared performance in Table 3, but the settings are very unclear.

(c). Why the method is described as an end-to-end approach? The surrogate modeling and control part are separate.

(d). Can the authors show some control curves, the state variables before and after the controls? This would help validate the efficacy of the proposed algorithm.

(e). Can the authors describe in more details for current methods, why the computation time are much longer than the proposed method? Is that due to the intractability of the formulated MPC problem, or due to the small simulation resolution, or due to the existence of constraints?

(f). How do the proposed method scale up to larger district networks?

(g). Are their any noises throughout the sensing, measuring, and control procedures? How do they affect the algorithm design?

---

> ### Author Response · Authors · 2024-11-25
> **Answers to Reviewer XnjS**
>
> *The authors express their gratitude to the reviewer for his valuable review and constructive comments.*
> Given all the concording reviews we received, the paper seems to need undergoing major changes which we are unfortunately unable to perform and process to update the paper in the given time.
>
> We did try however to answer some of the questions where our choices are justifiable:
>
> **(a). What do RY, AY mean in the dataset? Needs a bit more explanation.**
>
> The acronyms RY and AY were explained at the end of section 4.1. They stand for “Real Year” and “Augmented Year” respectively. The idea is that we augmented the training dataset based on a “Real” historical data.
>
> **(b). For the real data mentioned in the paper, is it possible to get the ground truth physical model? And based on such a model, can standard Model Predictive Control or optimization-based methods give exact solutions? I notice the authors compared performance in Table 3, but the settings are very unclear.**
>
> The ground truth physical model was indeed very briefly mentioned and a reference for a detailed explanation of the model was given. Using standard MPC would give similar or even more exact solutions but would require more computational time, that is the base idea of our comparison in Table 3 where we give the equivalent computational time that would have been required to do the same number of iterations (i.e., how many simulations to run) to optimize the system.
>
> **(c). Why the method is described as an end-to-end approach?**
>
> The surrogate modeling and control part are separate. The end-to-end character is schematized in Figure 1 where the surrogate model is used inside the control loop.
>
> **(d). Can the authors show some control curves, the state variables before and after the controls? This would help validate the efficacy of the proposed algorithm.
> (e). Can the authors describe in more details for current methods, why the computation time are much longer than the proposed method? Is that due to the intractability of the formulated MPC problem, or due to the small simulation resolution, or due to the existence of constraints?**
>
> While these questions/propositions are definitely relevant we were somehow constrained by the limits of the pages of the paper. The computation time for the current methods is much longer due to both small simulation resolution (required for solar loop) and heavy dynamical constraints (particularly the biomass boiler constraints, given in Appendix A.1)
>
> **(f). How does the proposed method scale up to larger district networks?**
>
> This would have been of great interest indeed, however real-world data for such systems are very rare, not to say non-existent as they are confidential and private. We thought about creating “fictional” systems (equivalent to “toy dataset”) but then we lose the whole “real impact” character of the paper and also the comparison to historical data becomes impossible.
>
> Best regards.

---

> > ### Comment · Reviewer_XnjS · 2024-11-26
> >
> > Thanks for your clarifications. Hope the paper will be improved in the future.

---

### Meta-Review · Area_Chair_JyKg · 2024-12-16

**Metareview:**

This paper addresses a specific application: the optimal control of district heating networks. The authors propose a two-step approach: first, they learn a GNN-based state-space model as a surrogate for system dynamics, and then apply a genetic algorithm to optimize the controller. Simulation results indicate that the proposed method improves both computation time and solution accuracy.

While the problem is of significant societal importance and the work is partially validated on real-world data, the paper offers limited innovation in machine learning. The methods employed are based on existing techniques, making the work better suited for application-focused engineering venues than a machine learning conference. As highlighted by reviewers, the authors could increase the scientific contribution to machine learning by performing more comparative and ablation studies to justify the usage of the genetic algorithm over alternative approaches, such as model-based reinforcement learning, model-free decision-making, or model predictive control (with reported performance).

Given these limitations and the absence of substantial scientific contributions to the field of machine learning, the paper is not ready for publication in its current form.

**Additional Comments On Reviewer Discussion:**

During the discussion, the authors noted the need for significant revisions to address the raised concerns but were unable to implement such changes within the available timeframe. They provided partial justifications for some methodological choices but did not sufficiently resolve the primary issues identified.

---

### Decision · Program_Chairs · 2025-01-22

Reject